# Determinants of anemia among pregnant women attending antenatal care in Horo Guduru Wollega Zone, West Ethiopia: Unmatched case-control study

**Birhanu Daba Tulu[1,2], Emiru Merdassa Atomssa[2], Hylemariam Mihiretie Mengist[3]***

**1** Nekemte Town Health office, Nekemte, Ethiopia, **2** Department of Public Health, College of Health Sciences, Wollega University, Nekemte, Ethiopia, **3** Department of Medical Laboratory Sciences, College of Health Sciences, Debre Markos University, Debre Markos, Ethiopia

* hylemariam@gmail.com

## Abstract

### Background

Anemia is a common clinical problem contributing to increased maternal and fetal morbidity and mortality during pregnancy. Anemia can be caused by different factors apart from known diseases. The main aim of this study was to identify determinants of anemia among pregnant women attending antenatal care at the public health facilities of Horo Guduru Wollega Zone, West Ethiopia, 2017.

### Methods

Health facility-based unmatched case-control study was conducted among 191 anemic and 382 non-anemic pregnant women from September 7, 2017, to October 25, 2017, in Horo Guduru Wollega Zone, West Ethiopia. Data were collected using pre-tested questionnaires from nine health facilities. Hemoglobin level determination, hemo-parasite diagnosis, venereal disease research laboratory (VDRL) test, and stool examination were done in the laboratories of the respective health centers. Cleaned and coded data were entered and analyzed using SPSS version 20. Frequency, proportion, mean and standard deviation were computed to summarize the data and presented by tables and bar graphs. Multivariate binary logistic regression analysis was used to determine the association of predictors and response variables at $P \le 0.05$. Adjusted odds ratio with 95% CI was used to show the strength of association between predictors and outcome variables.

### Results

A total of 573 pregnant women were enrolled in this study. Monthly income < 500 Ethiopian birr (AOR = 9.16, 95% CI: 4.23, 19.82), heavy menstrual bleeding (AOR = 2.38, 95%CI: 1.38, 4.09), taking iron supplement irregularly (AOR = 2.87, 95%CI:1.41, 5.84), Mid-upper Arm Circumference (MUAC) < 23 cm (AOR = 3.42, 95%CI: 2.07, 5.63), low dietary diversity score (AOR = 12.30, 95%CI: 4.64, 32.72), medium dietary diversity score (AOR = 3.40,

**Data Availability Statement:** All the data sets during and/or analyzed during the current study are available in the manuscript and supporting

documents. There is no restriction in the availability of data.

**Funding:** The author(s) received no specific funding for this work.

**Competing interests:** The authors have declared that no competing interests exist.

95%CI:1.48, 7.84) and intestinal helminthic infections (AOR = 6.31, 95%CI: 3.44, 11.58) were significantly associated with anemia during pregnancy.

## Conclusion

Average monthly income < 500 Ethiopian birr, heavy menstrual bleeding, low and medium dietary diversity score, taking of iron supplements irregularly, MUAC < 23 cm and intestinal helminthic infections were identified as independent determinants of anemia during pregnancy. Therefore, improving dietary diversity intake, routine deworming and empowering women on taking iron regularly are vital to prevent anemia during pregnancy.

## Introduction

Anemia implies a reduction in the oxygen-carrying capacity of the blood as a result of fewer circulating erythrocytes than normal or a decrease in the concentration of hemoglobin (Hb). Anemia during pregnancy is defined as a hemoglobin concentration less than 11gram per deciliter (g/dl) and classified as mild (10.0–10.9g/dl), moderate (7.0–9.9g/dl) and severe <7g/ dl. Currently, World Health Organization (WHO) recognized that the hemoglobin value less than 11.0 g/dl at 1st and 3rd trimesters and less than 10.5 g/dl in the 2nd trimester is used to define anemia [1].

Anemia is a common public health problem affecting one- third of the world's population. It is more common in women who are young, poor and pregnant [2]. It is among the most clinical problem contributing to increased maternal and fetal morbidity and mortality during pregnancy [3]. Anemia among pregnant women is more prevalent in developing countries than in developed countries [4].

The causes of anemia in pregnancy are multi-factorial. Iron folate, amino acids, vitamin A, C and other vitamin B complex deficiencies, intestinal helminthic infections, malaria, and chronic illnesses are among the most common causes [5]. The other risk factors include poverty, grand parity, too early pregnancies, too many children, frequent pregnancy spacing of less than one year, low socioeconomic status, illiteracy, late booking for antenatal care and gestational age [6, 7].

Globally, 56 million pregnant women were anemic and 20% of maternal deaths were caused by anemia. It was 35–75% among pregnant women in developing countries and 18% in developed countries. Africa (61.3%) and southeast Asia (52.5%) are regions with the highest rate of anemia during pregnancy in the world [8].

In Ethiopia, the prevalence of anemia among pregnant women was 22% which is more common in rural areas. According to the Central Statistical Agency of Ethiopia 2011 report [9], anemia is still a public health problem in the country. Anemia during pregnancy is widely distributed in different regions of Ethiopia especially in Somali (49.9%), Afar (40.4%) and Dire Dawa (33.2%).

Although a lot of researches have been conducted about anemia during pregnancy in Ethiopia, there is still a paucity of published data on determinants of anemia among pregnant women in the current study area. The study is; therefore, aimed to identify determinants of anemia among pregnant women in Horo Guduru Wollega Zone, West Ethiopia.

## Materials and methods

### Study setting and context

An institutional-based unmatched case-control study was conducted in randomly selected 8 health facilities. These facilities were Jima Rare health center, Goban health center, Kombolcha

health center, Ayale health center, Harato health center, Gudane health center, site 20 health center, site 24 health center, and Shambu hospital. The study was conducted from September 7, 2017, to October 25, 2017.

## Study population

Our study population was categorized into cases and controls. Cases were all pregnant women who were attending ANC follow up at the selected public health facilities whose Hb level fall in our anemia definition whereas controls were all pregnant women who were attending ANC follow up at the selected public health facilities whose Hb fall in our non-anemic definition. Anemia during pregnancy was defined and classified based on WHO [1] and Ethiopian ANC follow-up guidelines for different gestational ages. Accordingly, women in the first and third trimester whose Hb level was < 11g/dl and women, in the second trimester, whose Hb level was < 10.5 g/dl0 were defined to be anemic for cases. Pregnant women at any gestational age (GA) were included in the study. GA of pregnant women was classified into trimesters i.e. GA below 12 weeks: first trimester, GA of 13–24 weeks: second trimester and GA above 24 weeks: third trimester. The gestational age was calculated from last normal menstrual period (LNMP) which was determined by antenatal care providers. In Ethiopia, health extension workers register pregnant women of any GA with their last normal menstrual period at home, thus, GA is not usually missed.

Pregnant women who were taking anti-helminthic drugs within the past two weeks, very sick and unable to give information, those with confirmed acute and/ chronic disease-causing anemia and those under invasive anemia treatment except iron folate were excluded from this study.

## Sample size determination and sampling procedure

The sample size was calculated using OpenEpi software for unmatched case-control study by taking 95% confidence level, 80% power and a ratio of controls to cases 2:1 (r = 2), P1 = 42.03 and P2 = 57.97 by using OR = 1.9 [10]. Based on this, the intestinal parasitic infection was taken as the main exposure variable with a proportion of 57.97 among cases and 42.03% among controls with OR = 1.90 which gives a sample size of 347. Lastly, by considering a design effect of 1.5 and adding 10% non-response rate, the final sample size was 573. From this, cases were 191 and controls were 382. The study participants were consecutively enrolled until the planned sample size was achieved. A 2: 1 control to case ratio was applied and study participants were recruited proportionally from each health facilities based on the number of pregnant women attending ANC.

## Data collection

The questionnaire was developed from previous literatures and modified to the current context of the study based on the conceptual framework and study variables [6, 10 and 11]. The questionnaire was first prepared in English language and translated to 'Afan Oromo' which is the local language at the study sites and retranslated back to English language for consistency. Before the actual data collection, the questionnaire was pre-tested on 19 pregnant women (5% of the total sample size) who were attending ANC in an un-selected health facility, Shambu health center.

Minimum Dietary Diversity for Women (MDD-W) was calculated from 24 hours' dietary recall data. All foods consumed day before the study were categorized into 10 food groups. Consuming a food item from any of the groups was assigned as" Yes" and if not consumed assigned as" No". Dietary diversity score of 10 points was computed by adding all values of the

groups. Then, the added values were categorized as low (≤3), medium (4–6) and high (≥7) dietary intake [12]. Study participants were screened for nutritional status by measuring MUAC (Mid-Upper Arm Circumference) by using tape meter.

## Specimen collection and processing

Blood samples were collected following aseptic procedures for hemoglobin determination using microhematocrit centrifugation technique. Briefly, blood was collected into two-third of the micro-hematocrit tube with anti-coagulant and centrifuged for 5 minutes and then hemoglobin was determined by dividing hematocrit value by three [13]. The hemoglobin cut-off value for anemia during pregnancy was determined according to the WHO [1] and Ethiopian ANC follow-up guidelines. Thick and thin blood films were made from collected blood samples and stained by Giemsa staining technique for hemo-parasite diagnosis [14] and VDRL test [15] was done to identify Syphilis infection following standard operating procedures. Further, formalin-ether concentration technique was used for diagnosing ova and larvae of helminths [14].

## Data quality control

One senior midwife who was working in ANC department of each selected health facilities and three Health Officers were recruited as supervisors who were under the whole supervision of the principal investigator. Two days of training was conducted for data collectors and supervisors to aid for clarity, consistency, and reliability of measurements. Collected data were checked for consistency and completeness daily and pre-analytical, analytical and post-analytical laboratory data quality were controlled accordingly.

## Data analysis

Data were cleaned, coded and entered into SPSS software version 20 for analysis. Outcome variables were dichotomized into 1 = case and 0 = control. Binary logistic regression models were used to determine the association between predictors and outcome variables. Bivariate analysis was carried out for each independent variable to identify the presence of association with response variable at $P < 0.25$ [16]. Multivariate logistic regression was used to control the possible confounding factors at $P \leq 0.05$. Adjusted odds ratio with 95% confidence (AOR, 95%CI) was used to infer the results and all data analyses were done according to standard protocols [17–19].

## Ethics considerations

The study was conducted after it was ethically reviewed and approved by the Research and Ethical Review Committee of Wollega University. Then, an official letter of co-operation was written to selected health facilities and permission was obtained. The ethical review committee approved informed verbal consent documented by a witness after the objectives had been explained. Participants' right during the interview for either not to participate or to withdraw at any stage of the interview was guaranteed. All the information obtained from the study participants were coded to maintain confidentiality and positive results were immediately communicated with clinicians for appropriate intervention.

# Results

## Socio-demographic characteristics of study participants

A total of 573 pregnant women participated in the study. The mean age of respondents was 27 (±5.4) years in cases and 26.8 (±5) years in controls. More than half of the participants, (59.2%

**Table 1. Sociodemographic characteristics of study participants who were attending ANC in the public health facilities of Horo Guduru Wollega Zone, West Ethiopia from September 7 to October 25, 2017 (n = 573).**

| Variables | | Frequencies (%) | |
|---|---|---|---|
| | | Cases | Controls |
| Age | 15–19 | 6(3.1) | 20(5.2) |
| | 20–24 | 52(27.2) | 91(23.8) |
| | 25–29 | 67(35.1) | 152(39.8) |
| | 30–34 | 40(21.1) | 78(13.6) |
| | 35–39 | 21(11) | 38(10) |
| | 40–44 | 5(2.6) | 3(0.8) |
| Marital status | Never married | 3(1.6) | 2(0.5) |
| | Married | 181(94.8) | 369(96.6) |
| | Widowed | 4(2.1) | 4(1) |
| | Divorced | 3(1.)6 | 7(1.8) |
| Residence | Urban | 73(38.2) | 168(44) |
| | Rural | 118(61.8) | 214(56) |
| Family size | <5 | 115(60.2) | 240(62.8) |
| | ≥5 | 76(40) | 142(37.2) |
| Occupation of respondent | Governmental employee | 26(13.6) | 77(20.2) |
| | Farmer | 113(59.2) | 201(52.6) |
| | Merchant | 32(16.8) | 58(15.2) |
| | Daily laborer | 11(5.8) | 14(3.7) |
| | Other | 9(4.7) | 32(8.4) |
| Occupation of husband | Governmental employee | 39(20.4) | 98(25.7) |
| | Farmer | 115(60) | 199(52) |
| | Merchant | 20(10.5) | 55(14.4) |
| | Daily laborer | 13(6.8) | 23(6) |
| | Other | 4(2.1) | 7(1.8) |
| Income | <500 | 85(44.5) | 40(10.5) |
| | 500–1499 | 53(27.7) | 146(38.2) |
| | 1500–2499 | 22(11.5) | 54(14) |
| | 2500–3499 | 13(6.8) | 39(10.2) |
| | >3500 | 18(9.4) | 103(27) |
| Educational status | No education | 59(31) | 74(19.3) |
| | Primary school | 75(39.3) | 151(39.5) |
| | Secondary school | 24(12.6) | 63(16.5) |
| | Above secondary school | 33(17.3) | 94(24.6) |

(113/191) of cases and 52.6% (201/382) of controls, were farmers. The proportion of anemic pregnant mothers, 44.5% (85/191) with average monthly income < 500 ETB were more than four times of their counterparts, 10.5% (40/382) (**Table 1**).

## Maternal obstetric factors

Most of the study participants, 98% (185/188) of cases and 96.3(366/380) of controls, were married with a mean age of 19(± 2.3). Clinically, 63.9% (122/191) of cases and 63% (241/382) of controls were at their third trimester. The proportion of heavy menstrual bleeding (using more than 7 tampons per menstrual period) was two times more in cases than in controls (32% and 16%, respectively) (**Table A in S1 File**).

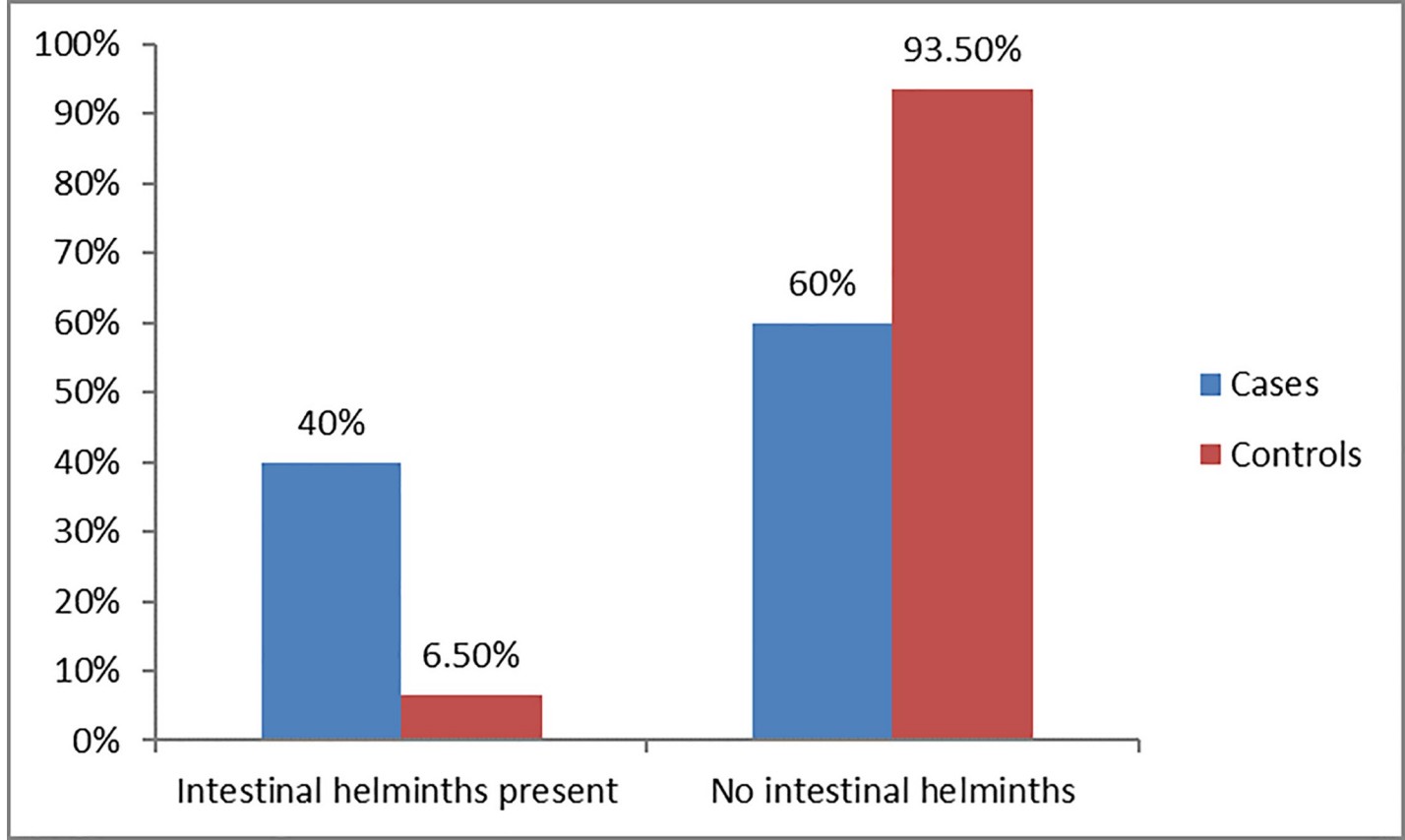

**Fig 1. Distribution of intestinal helminthic infection among pregnant women.**

### Maternal dietary factors

Based on the dietary diversity score, more than half of cases (66% (126/191)) and controls (67.3%(257/382)) were classified under medium dietary diversity score, whereas 4.2% (8/191) and 30% (57/191) of cases, and 26%(99/382) and 6.8%(26/382) of controls were classified under high and low dietary diversity scores, respectively (**Table B in S1 File**).

### Infection related factors

The proportion of malaria infection was almost twelve times in cases, 6% (11/191) than controls, 0.5% (2/382). VDRL test was positive in 3.7% (14/191) of cases and 10% (19/382) of controls. Intestinal helminthic infections were diagnosed in 39.9% (76/191) of cases and 6.5% (25/382) of controls (**Fig 1** and **Table 2**).

### Hygiene and sanitation related factors

Majority of the study participants i.e. 87.4% (167/191) of cases and 91.9% (350/382) of Controls utilized latrine and 70.2% (134/191) cases and 64.4% (246/382) controls used to dispose of solid waste in open field. The source of drinking water was tap water for 45.5% (87/191) of cases and 52.6% (201/382) of controls (**Table 3**).

**Table 2. Diseases related factors of anemia among pregnant women who were attending ANC follow up in the selected public health facilities of Horo Guduru Wollega Zone, West Ethiopia from September 7 to October 25, 2017(n = 573).**

| Variables | | | Frequencies (%) | |
|---|---|---|---|---|
| | | | Cases | Controls |
| Malaria | Yes | | 11(5.8) | 2(0.5) |
| | No | | 180(94.2) | 380(99.5) |
| Diarrhea in the last 2 weeks | Yes | | 22(11.5) | 21(5.5) |
| | No | | 169(88.5) | 361(94.5) |
| Syphilis infection | Yes | | 19(10) | 14(3.7) |
| | No | | 172 (90) | 368(96.3) |
| Intestinalhelminthicinfection | Yes | | 76(39.9) | 25(6.5) |
| | No | | 115(60.1) | 357(93.5) |
| | *Ascaris Lumbricoides* | Yes | 29(15) | 12(6.3) |
| | | No | 161(84.7) | 371(96.8) |
| | *Tricuris Trichiura* | Yes | 14(7.4) | 5(1.3) |
| | | No | 176(92.6) | 378(98.7) |
| | *Hookworm* | Yes | 27(14) | 6(1.56) |
| | | No | 163(85.8) | 377(98.4) |
| | *Hymenolepis nana* | Yes | 8(4.2) | 4(1) |
| | | No | 182(95.8) | 379(98.9) |

## Status of anemia

Mild anemia was the most prevalent form of anemia accounting for 60% followed by moderate anemia 39% and severe anemia 1%. The mean hemoglobin level was 9.7 ± 0.69 gm/dl among the cases and 13 ± 1.1 gm/dl among control groups.

**Table 3. Hygiene and sanitation-related factors of anemia in pregnant women attended ANC follow up in the selected public health facilities of Horo Guduru Wollega Zone, West Ethiopia from September 7 to October 25, 2017 (n = 573).**

| Variables | | Frequencies (%) | |
|---|---|---|---|
| | | Cases | Controls |
| Using latrine | Yes | 167(87.4) | 351(91.9) |
| | No | 24(12.6) | 31(8.1) |
| Hand washing facility with latrine | Yes | 36(18.8) | 104(27.2) |
| | No | 139(72.8) | 264(72.8) |
| Solid waste disposal site | Open-pit | 41(21.5) | 90(23.6) |
| | Open field | 134(70.2) | 246(64.4) |
| | Sanitary landfill | 15(7.4) | 46(12) |
| Source of drinking water | Tap water | 87(45.5) | 201(52.6) |
| | Protected spring | 56(29.3) | 115(30) |
| | Unprotected spring | 32(16.8) | 42(11) |
| | River | 14(7.3) | 20(5.2) |
| | Other | 2(1) | 4(1) |
| Treatment for the not potable water source | Yes | 78(75) | 150(82.8) |
| | No | 79(76) | 137(75.6) |
| Handwashing before food preparation | Yes | 190(99) | 381(96.9) |
| | No | 1(0.5) | 1(0.3) |

## Inferential statistics

Multivariable binary logistic regression analysis was done to identify independent predictors of anemia at $P \leq 0.05$. Hosmer and Lemeshow goodness of model fit test showed a good model fit ($P = 0.98$) and the VIF of covariates in this study ranged between 1–1.2 which is in the acceptable range for multicollinearity which means there was no correlation among predictors of anemia and; therefore, there was no correlation between gestational age and other risk factors of anemia. This indicates that risk factors of anemia were not changeable based on differences in gestational age.

Average monthly income < 500 ETB showed statistically significant association with anemia among pregnant women (AOR = 9.16, 95% CI: 4.23, 19.82). Anemia was more prevalent among those with heavy menstrual bleeding before index pregnancy than their counterparts (AOR = 2.38, 95%CI: 1.38, 4.09). Pregnant women who took iron supplements irregularly were almost 3 times more likely to be anemic than their counterparts (AOR = 2.87, 95%CI: 1.41, 5.84). A strong association was also seen between the occurrence of anemia and low dietary diversity score of 24 hours recall (AOR = 12.3, 95%CI: 4.64, 32.72). Likewise, there was also a significant association between medium DDS and occurrence of anemia among pregnant women (AOR = 3.40, 95%CI: 1.48, 7.84). Gestational age was not significant predictor of anemia (P > 0.05) (**Table 4**).

Anemia was almost 6 times more common among pregnant women who had intestinal helminthic infection than those with no intestinal helminthic infection (AOR = 6.31, 95%CI: 3.44, 11.58). Pregnant women infected with *Ascaris Lumbricoides*, *Tricuris Trichiura*, Hookworm and *Hymenolepis nana* were significantly affected by anemia (AOR = 6.81, 95%CI: 3.35, 13.85, AOR = 8.12, 95% CI: 2.85, 23.16, AOR = 13.03, 95%CI: 5.24, 32.45, AOR = 4.88, 95%CI: 1.38, 17.14), respectively (**Table 4**).

## Discussion

Anemia is one of the most public health important diseases among pregnant women in Ethiopia [9] and in developing countries in general [20]. Anemia has multifactorial causes and the risk factors are different across populations. Here we identified six determinants of anemia among pregnant women who were attending ANC in the specific study sites.

More than half of anemic pregnant women had a mild type of anemia which was followed by moderate and severe anemia in this study. This is comparative with the studies conducted in Asosa zone, Gilgel Gibe dam area, and East Wollega Zone, Ethiopia [21, 22 and 23]. But, it is not comparative with the studies conducted in Tikur Anbessa Specialized hospital, Wolayita Sodo town and Mekelle town, Ethiopia [5, 10, and 24]. This difference might be due to the difference in determinants of anemia from one geographic area to another geographical area and sample size of the studies. Further, the differences in study design could also contribute to the observed inconsistencies.

An average monthly income of the household was identified as a significant predictor of anemia among pregnant women in this study. This finding is consistent with another study conducted in Azezo health center and Sidama Zone, Ethiopia [6, 25]. In contrast, this finding is different from a study done at Nekemte health center, Gilgel Gibe dam area, Buta Jira General Hospital, and Dera district, Ethiopia [11, 22, 26 and 27]. This difference could possibly be due to the difference in the economic status among the participants as low family income leads to food insecurity with the consequence of malnutrition which in turn leads to iron deficiency anemia [28].

Heavy menstrual bleeding (using more than seven sanitary pads per menstrual period) before the current pregnancy was identified as one independent factor for the occurrence of

**Table 4. Multivariate binary logistic regression analysis of determinants of anemia among pregnant women in the selected public health facilities of Horo Guduru Wollega Zone, West Ethiopia from September 7 to October 25, 2017(n = 573).**

| Variables | | Frequencies(%) | | COR(95%CI) | AOR(95%CI) | P value |
|---|---|---|---|---|---|---|
| | | Cases | Controls | | | |
| Income | <500 | 85(44.5) | 40(10.5) | 12.16(6.5,22.74)* | 9.16(4.23,19.82) | <0.001*** |
| | 500–1499 | 53(27.7) | 146(38.2) | 2.08(1.15, 3.75)* | 1.83(0.88, 3.81) | 0.11 |
| | 1500–2499 | 22(11.5) | 54(14) | 2.33(1.15, 4.72)* | 1.83(0.78, 4.29) | 0.17 |
| | 2500–3499 | 13(6.8) | 39(10.2) | 1.91(0.86, 4.26) | 2.53(0.97, 6.63) | 0.06 |
| | >3500 | 18(9.4) | 103(27) | 1 | 1 | |
| Menstrual period | ≥8 days | 46(24) | 32(8.4) | 3.5(2.12, 5.67)* | 2.11(0.97, 4.59) | 0.06 |
| | <8 days | 145(76) | 350(91.6) | 1 | 1 | |
| Heavy menstrual bleeding | Yes | 61(31.9) | 61(16) | 2.47(1.64, 3.72)* | 2.38(1.38, 4.09) | 0.002** |
| | No | 130(68.4) | 321(83.8) | 1 | 1 | |
| Parity | Nulliparous | 53(28) | 132(34.5) | 1 | 1 | |
| | Primipara | 40(20.9) | 80(20.9) | 1.24(0.76, 2.04) | 2.52(0.99, 6.44) | 0.054 |
| | Multipara | 45(23.7) | 92(24) | 1.22(0.76,1.97) | 1.82(0.72, 4.59) | 0.20 |
| | Grandpara | 53(28) | 78(20.4) | 1.69(1.06, 2.72)* | 2.28(0.91, 5.74) | 0.08 |
| Abortion | Yes | 28(14.7) | 34(8.9) | 1.76(1.03, 2.99)* | 0.92(0.44, 1.94) | 0.83 |
| | No | 163(85.3) | 348(91) | 1 | 1 | |
| Birth interval | <2 years | 26(13.7) | 21(5.5) | 2.87(1.55, 5.30)* | 2.22(0.85, 5.76) | 0.10 |
| | ≥2 years | 116(61) | 272(71) | 1 | 1 | |
| ANC follow up | Yes | 147(77) | 350(91.6) | 1 | 1 | |
| | No | 44(23) | 32(8.4) | 3.27(1.99, 5.37)* | 1.99(0.75, 5.31) | 0.17 |
| Trimester | First | 12(6.3) | 42(11) | 1 | | |
| | Second | 55(29) | 120(31.3) | 1.47(0.96, 2.24) | | |
| | Third | 123(64.7) | 221(57.7) | 1.41(0.79, 2.51) | | |
| Bleeding | Yes | 22(11.6) | 13(3.4) | 3.7(1.82,7.51)* | 1.25(0.41, 3.76) | 0.69 |
| | No | 169(88.5) | 369(96.6) | 1 | | |
| Nausea and vomiting | Yes | 63(33) | 89(23.3) | 1.62(1.10, 2.38)* | 0.98(0.54, 1.82) | 0.97 |
| | No | 128(67) | 293(76.7) | 1 | 1 | |
| Took iron | Yes | 104(54.5) | 273(71.5) | 1 | 1 | |
| | No | 87(45.8) | 109(28.5) | 2.09(1.46, 3.01)* | 1.29(0.69, 2.39) | 0.43 |
| Taking iron supplement daily | Yes | 63(60.6) | 239(87.6) | 1 | 1 | 1 |
| | No | 41(39.4) | 34(12.4) | 4.58(2.69, 7.79)* | 2.87(1.41, 5.84) | 0.004** |
| Frequency of eating vegetables | ≤3/week | 56(29.3) | 175(45.8) | 1 | 1 | |
| | ≥3/week | 135(70.7) | 207(54.2) | 2.04(1.41, 2.95)* | 1.33(0.72, 2.47) | 0.36 |
| MUAC | <23 | 96(50.3) | 58(15.2) | 5.72(3.84, 8.53)* | 3.42(2.07, 5.63) | <0.001** |
| | ≥23 | 95(49.7) | 324(84.8) | 1 | 1 | * |
| DDS | Low | 57(30) | 26(6.8) | 27.13(11.5,63.9)* | 12.3(4.64, 32.72) | <0.001** |
| | Medium | 126(66) | 257(67.3) | 6(2.86,12.86)* | 3.40(1.48, 7.84) | * |
| | High | 8(4.2) | 99(25.9) | 1 | 1 | 0.004** |
| *Ascaris Lumbricoides* | Yes | 29(15.2) | 12(3.14) | 5.57(2.7, 11.19)* | 6.81(3.35,13.85) | <0.001*** |
| | No | 47(24.6) | 13(3.4) | 1 | 1 | |
| *Tricuris Trichiura* | Yes | 14(7.3) | 5(1.3) | 6.01(2.13,16.96* | 8.12(2.85, 23.16) | <0.001*** |
| | No | 176(92) | 378(98.9) | 1 | 1 | |
| *Hookworm* | Yes | 27(14) | 6(1.57) | 4.17(1.24, 14.0)* | 13(5.24,32.45) | <0.001*** |
| | No | 49(25.6) | 19(4.97) | 1 | 1 | |

*(Continued)*

**Table 4.** (Continued)

| Variables | | Frequencies(%) | | COR(95%CI) | AOR(95%CI) | P value |
|---|---|---|---|---|---|---|
| | | Cases | Controls | | | |
| *Hymenolepis nana* | Yes | 8(4.2) | 4(1) | 10.41(4.22, 25.7) | 4.88(1.38, 17.14) | 0.01* |
| | No | 68(35.6) | 21(5.49) | 1 | 1 | |

*P < 0.05

**P< 0.01

***P< 0.001

MUAC: Mid-upper arm circumference, DDS: Dietary diversity score

anemia among pregnant women. Anemic pregnant women who had heavy menstrual bleeding were almost twice of their counterparts in this study. This finding is in agreement with a study conducted at the Federal Medical Center Owemi, Nigeria [29], Otona Hospital, Wolayita Soddo town and Buta Jira hospital, Ethiopia [10, 26]. Excess blood flow leads to iron storage depletion and causes iron deficiency anemia which commonly occurs in pregnant women with short interbirth intervals.

Irregular use of iron supplements was identified as one of the determinants of anemia in the current study which is comparative with reports from East Wollega, Ethiopia [23], Kathmandu tertiary level hospital in the United States [30] and Pumwani maternity hospital in Kenya [31]. Regular use of iron supplements is vital to prevent anemia as the demand for iron is higher due to the increased blood supply during pregnancy.

Pregnant women with mid-upper arm circumference (MUAC) less than 23 cm (malnourished) were more at risk of being anemic. This finding is in line with the results of the studies conducted in Kenya [31] and other parts of Ethiopia like Gilgel Gibe dam area, Dessie town, and urban areas of east Ethiopia [22, 32, and 33]. Measuring MUAC is a routine activity for screening malnutrition during ANC follow up in Ethiopia. Malnutrition is one of the commonest contributing factors for iron deficiency anemia.

Pregnant women with low (≤3) and medium (4–6) dietary diversity score were more at risk of developing anemia when compared with those with dietary diversity score >6. This finding was supported by a study conducted in Mekele town, Ethiopia [24] and another study conducted from South Ethiopia [34]. Minimum dietary diversity score (MDDS) is used as the proxy measure of nutritional quality of pregnant women. Since pregnancy is a period demanding physiologically high nutrition, it is advisable to diversify the diet than the usual one. Low average monthly income and food taboos during pregnancy in rural communities might cause this low and medium dietary diversity scores in the study area which in turn result in anemia.

Consistent with previous studies conducted in different regions of the world; Kenya [31], India [35] and Nigeria [36], and other parts of Ethiopia [10, 23, 29,37], intestinal helminthic infection was one strong driving factor of anemia during pregnancy in the current study. *Ascaris Lumbricoides* and Hookworms were the predominant intestinal helminths infections among pregnant women in the current study. This finding is comparable with the studies conducted in East Wollega zone and Lemo district, Ethiopia [23, 34]. Intestinal helminths cause loss of appetite, reduce absorption, bleeding and share food uptake which in turn leads to iron deficiency anemia. Since the majority of the study participants were farmers and rural residents, they had a high chance of exposure to these intestinal helminthic infections due to walking barefooted and poor hygiene of food and water.

In contrary to the current study, variables like age, residence, family size, ANC follow up, trimester, malaria infection and interbirth interval showed statistically significant association

with anemia among pregnant women in the previous studies conducted in Ethiopia [10, 21, 32, 33, 38 and 39]. This difference might be due to the fact that determinants of anemia varied from one geographical area to another geographical area due to the differences in the nutritional, altitudinal, economic and educational status of the study participants.

The study was facility-based which couldn't be generalized to all pregnant women left in the community (didn't attend antenatal care follow up). Dietary diversity score of 24 hours recall might leads to recall bias and the amount of diet also couldn't be measured. There might also be a social desirability problem to explain income level and some foods that were seen as low food status within the community. Having these limitations, we believe that the results of this study can contribute to policymakers and clinicians to prevent anemia during pregnancy.

## Conclusion

This study identified determinants of anemia among pregnant women in the study area. Average monthly income < 500 ETB, heavy menstrual bleeding before the index pregnancy, low and medium dietary diversity score, MUAC < 23 cm, taking iron supplementation irregularly and intestinal helminthic infections were identified as independent predictors of anemia. Early identification of determinants of anemia and appropriate interventions have paramount importance in fighting anemia to help mothers enjoy their pregnancy. Strengthening health education and counseling on diversifying dietary intake, increasing inter birth interval, routine stool examination, de-worming, and hygiene are valuable. Further large scale community-based studies are recommended.

## Supporting information

**S1 File. Supporting information 1: Supporting Tables Table A and Table B.**
(DOCX)

**S2 File. Supporting information 2: Questionnaires used to collect data from pregnant women.**
(DOCX)

## Acknowledgments

We would like to acknowledge Wollega University administrative assistance to conduct this research. Lastly, but not the least, our special thanks and sincere appreciation also go to all data collectors, supervisors and study participants without whom the research would not be a reality.

## Author Contributions

**Conceptualization:** Birhanu Daba Tulu, Emiru Merdassa Atomssa, Hylemariam Mihiretie Mengist.

**Data curation:** Birhanu Daba Tulu.

**Formal analysis:** Birhanu Daba Tulu, Hylemariam Mihiretie Mengist.

**Investigation:** Birhanu Daba Tulu.

**Methodology:** Birhanu Daba Tulu, Emiru Merdassa Atomssa, Hylemariam Mihiretie Mengist.

**Project administration:** Birhanu Daba Tulu, Hylemariam Mihiretie Mengist.

**Resources:** Birhanu Daba Tulu, Emiru Merdassa Atomssa, Hylemariam Mihiretie Mengist.

**Software:** Birhanu Daba Tulu, Emiru Merdassa Atomssa, Hylemariam Mihiretie Mengist.

**Supervision:** Emiru Merdassa Atomssa, Hylemariam Mihiretie Mengist.

**Validation:** Birhanu Daba Tulu.

**Visualization:** Birhanu Daba Tulu.

**Writing – original draft:** Birhanu Daba Tulu, Emiru Merdassa Atomssa, Hylemariam Mihiretie Mengist.

**Writing – review & editing:** Emiru Merdassa Atomssa, Hylemariam Mihiretie Mengist.

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
