## [Decision Letter · Decision Letter 0]

31 Jul 2019

PONE-D-19-15457

Determinants of Anemia among Pregnant Women Attending Antenatal Care in Horo Guduru Wollega Zone, West Ethiopia: Unmatched case-control study

PLOS ONE

Dear  Mr. Mengist,

Thank you for submitting your manuscript to PLOS ONE. After careful consideration, we feel that it has merit but does not fully meet PLOS ONE’s publication criteria as it currently stands. Therefore, we invite you to submit a revised version of the manuscript that addresses the points raised during the review process. 

The manuscript sounds like a replication of work done in other districts of Ethiopia.Literature review should be expanded to include work done outside Ethiopia. Your English should be edited. Make a conscious effort to read and correct all the grammatical errors in the manuscript.

We would appreciate receiving your revised manuscript by August 14, 2019.. To enhance the reproducibility of your results, we recommend that if applicable you deposit your laboratory protocols in protocols.io, where a protocol can be assigned its own identifier (DOI) such that it can be cited independently in the future. For instructions see: http://journals.plos.org/plosone/s/submission-guidelines#loc-laboratory-protocols

We look forward to receiving your revised manuscript.

Kind regards,

Mary Glover-Amengor

Academic Editor

PLOS ONE

Journal Requirements:

3. Please include additional information regarding the survey or questionnaire used in the study and ensure that you have provided sufficient details that others could replicate the analyses. For instance, if you developed a questionnaire as part of this study and it is not under a copyright more restrictive than CC-BY, please include a copy, in both the original language and English, as Supporting Information. Moreover, please include more details on how the questionnaire was pre-tested, and whether it was validated.

4. Please correct your reference to "p=0.000" to "p<0.001" or as similarly appropriate, as p values cannot equal zero.

Reviewers' comments:

Reviewer's Responses to Questions

**Comments to the Author**

1. Is the manuscript technically sound, and do the data support the conclusions?

Reviewer #1: Partly

2. Has the statistical analysis been performed appropriately and rigorously? 

Reviewer #1: No

3. Have the authors made all data underlying the findings in their manuscript fully available?

Reviewer #1: Yes

4. Is the manuscript presented in an intelligible fashion and written in standard English?

Reviewer #1: No

5. Review Comments to the Author

Reviewer #1: This article is on the determinants for anemia in pregnant women in Ethiopia.

The study design includes a case-control study.

This article includes some limitations.

Women were recruited at ANC visit. The inclusion criteria were unclear especially regarding which ANC visit. What was the gestational age at inclusion? How was gestational age assessed? These are important criterias to mention and to discuss: Hb changes over the course of pregnancy (Ouedraogo et al.). Different cut-offs have been proposed according to the timing in pregnancy. This is not explained nor discussed, although this introduces some change in women included or not in the case/control group.

Also, what are guidelines in Ethiopia regarding anemia, malaria, iron/folic acid supplements, treatments for helminths, and their timing of administration? That would influence the level of anemia according to gestational age, and risk factors according to gestational age.

Some parts are too detailed for a scientific article. For example, data quality control, SOPs, etc. As well as some parts of data analysis.

100% response rate: this is strange. All women accepted to participate?

Severity of anemia should be described.

The language should be revised.

The results should refer to the tables explicitly. The p-value should be added in the tables were appropriate.

Some abbreviations in the tables are not explained (for example, MUAC, DDS).

Intestinal helminthic infections and the type of helminths infection are both included in the multivariate analyses, as presented. However, they are highly correlated, and it makes no sense to include both of them.

The number of tables should be reduced.

6. PLOS authors have the option to publish the peer review history of their article (what does this mean?). If published, this will include your full peer review and any attached files.

Reviewer #1: No

---

## [Author Response · Author response to Decision Letter 0]

13 Aug 2019

Dear editor and reviewers

PONE-D-19-15457

Determinants of Anemia among Pregnant Women Attending Antenatal Care in Horo Guduru Wollega Zone, West Ethiopia: Unmatched case-control study 

First of all, we would like to thank you for providing the reviewers’ comments which are very constructive and crucial. We have incorporated all the point by point below. We have formatted the manuscript based on the PLOSONE guidelines. We have edited the English language online since we cannot afford to pay for editing agents. We have cleaned the manuscript as well. We have marked “red” the changes made in the manuscript and we have a separate file for “manuscript with track changes” and “clean manuscript”. We have also attached a “rebuttal letter” which responds editor and reviewers’ comments and/suggestions point by point. Further another document is also added as additional material to minimize the number of tables.

Response to editor

1. We have included literatures from abroad and thus our manuscript currently does not look a replication of previous works done in Ethiopia. 

2. We have copyedit manuscript for language usage, spelling, and grammar. The language in the manuscript looks better than it was before 

3. We have included additional information regarding the survey or questionnaire used in the study and we ensure that we have provided sufficient details that others could replicate the analyses. The validation and pre-test regarding the questionnaire is included in the manuscript.

4. We have corrected the P value reference to "p<0.001" 

5. We have changed our declaration regarding data availability to “All data and supporting documents are available freely without restriction”. We have added questionnaires used freely. 

Response to the reviewers

1. Women were recruited at ANC visit. The inclusion criteria were unclear especially regarding which ANC visit. What was the gestational age at inclusion? How was gestational age assessed? These are important criterias to mention and to discuss: Hb changes over the course of pregnancy (Ouedraogo et al.). Different cut-offs have been proposed according to the timing in pregnancy. This is not explained nor discussed, although this introduces some change in women included or not in the case/control group. 

Response: we have included clear inclusion criteria in the manuscript. The gestational age was assessed by the participants’ record books and patients’ response. The women were recruited in any ANC visit with any gestational age. Anemia definition cut-off value was categorized based on gestational age in both cases and controls as clearly described in the manuscript.

2. Also, what are guidelines in Ethiopia regarding anemia, malaria, iron/folic acid supplements, treatments for helminths, and their timing of administration? That would influence the level of anemia according to gestational age, and risk factors according to gestational age.

Response: We used WHO and Ethiopian ANC follow up guideline to define anemia in pregnant women Factors like malaria, iron/Folic acid supplements and deworming prophylaxis administration status those included on anemia guideline in Ethiopia were incorporated in the study.We have used Ethiopian ANC follow-up guideline to analyze our data regarding anemia and other factors

3. Some parts are too detailed for a scientific article. For example, data quality control, SOPs, etc. As well as some parts of data analysis. 100% response rate: this is strange. All women accepted to participate? Severity of anemia should be described. The language should be revised.

Response: We have briefly summarized the detailed parts like data quality control and SOPs and the data analysis. The response rate was 100% since we used consecutive sampling technique. We enrolled pregnant women consecutively until the sample size was reached. Thus we have removed the phrase “100% response rate”. Severity of anemia was described as Mild, moderate and severe in the manuscript. We have edited the language well.

4. The results should refer to the tables explicitly. The p-value should be added in the tables were appropriate. Some abbreviations in the tables are not explained (for example, MUAC, DDS). 

Response: The results refer the tables clearly. All P-values were added in the tables where appropriate and all abbreviations are explained.

5. Intestinal helminthic infections and the type of helminths infection are both included in the multivariate analyses, as presented. However, they are highly correlated, and it makes no sense to include both of them. The number of tables should be reduced.

 Response: We have removed the “intestinal helminthic infection” from the multivariate analysis. We have reduced the number of tables from 6 to 4 tables. We separately submitted the removed tables as “Supporting Table ST1”

---

## [Decision Letter · Decision Letter 1]

4 Oct 2019

PONE-D-19-15457R1

Determinants of Anemia among Pregnant Women Attending Antenatal Care in Horo Guduru Wollega Zone, West Ethiopia: Unmatched case-control study

PLOS ONE

Dear Mr Mengist,

Thank you for submitting your manuscript to PLOS ONE. After careful consideration, we feel that it has merit but does not fully meet PLOS ONE’s publication criteria as it currently stands. Therefore, we invite you to submit a revised version of the manuscript that addresses the points raised during the review process.

We would appreciate receiving your revised manuscript by Nov 17 2019 11:59PM. To enhance the reproducibility of your results, we recommend that if applicable you deposit your laboratory protocols in protocols.io, where a protocol can be assigned its own identifier (DOI) such that it can be cited independently in the future. For instructions see: http://journals.plos.org/plosone/s/submission-guidelines#loc-laboratory-protocols

We look forward to receiving your revised manuscript.

Kind regards,

Carmen Melatti

Academic Editor

PLOS ONE

on behalf of 

Mary Glover-Amengor

Academic Editor

PLOS ONE

Reviewers' comments:

Reviewer's Responses to Questions

**Comments to the Author**

1. If the authors have adequately addressed your comments raised in a previous round of review and you feel that this manuscript is now acceptable for publication, you may indicate that here to bypass the “Comments to the Author” section, enter your conflict of interest statement in the “Confidential to Editor” section, and submit your "Accept" recommendation.

Reviewer #1: (No Response)

2. Is the manuscript technically sound, and do the data support the conclusions?

Reviewer #1: Yes

3. Has the statistical analysis been performed appropriately and rigorously? 

Reviewer #1: No

4. Have the authors made all data underlying the findings in their manuscript fully available?

Reviewer #1: Yes

5. Is the manuscript presented in an intelligible fashion and written in standard English?

Reviewer #1: Yes

6. Review Comments to the Author

Reviewer #1: Thank you for Editing your manuscript.

There is still a major comment which is gestational age at enrolment. Gestational age at enrolment should be described. Indeed, the definition of anemia changes over pregnancy, and iron supplementts, anti-helminthic treatments, IPTp, etc are prescribed to the women during pregnancy. Thus, depending on the timing during pregnancy of enrolment in the study, risk factors may change. This should be discussed in the manuscript.

7. PLOS authors have the option to publish the peer review history of their article (what does this mean?). If published, this will include your full peer review and any attached files.

Reviewer #1: No

---

## [Author Response · Author response to Decision Letter 1]

4 Oct 2019

Dear editor and reviewers

PONE-D-19-15457R1

Determinants of Anemia among Pregnant Women Attending Antenatal Care in Horo Guduru Wollega Zone, West Ethiopia: Unmatched case-control study 

First of all, we would like to thank you for providing the reviewers’ comments which are very constructive and crucial. We have incorporated all the point by point below. We have highlighted “red” the changes made in the manuscript and we have a separate file for “manuscript with track changes” and a clean manuscript “manuscript”. We have also attached a rebuttal letter “Response to reviewers” which responds reviewers’ comments and/suggestions point by point. 

Response to the reviewers

1. There is still a major comment which is gestational age at enrolment. Gestational age at enrolment should be described. Indeed, the definition of anemia changes over pregnancy, and iron supplementts, anti-helminthic treatments, IPTp, etc are prescribed to the women during pregnancy. Thus, depending on the timing during pregnancy of enrolment in the study, risk factors may change. This should be discussed in the manuscript.

Response: Pregnant women at any gestational age (GA) were included in the study. GA of pregnant women was classified into trimesters i.e. GA below 12 weeks: first trimester, GA of 13- 24 weeks: second trimester and GA above 24 weeks: third trimester. The gestational age was calculated from last normal menstrual period (LNMP) which was determined by antenatal care providers. In Ethiopia, health extension workers register pregnant women of any GA with their last normal menstrual period at home, thus, GA is not usually missed. The proportion of gestational age among the study participants is already described in the manuscript and supporting files.

GA was not a significant predictor of anemia among cases and multicollinearity test was done by using VIF to assess presence of collinearity among independent variables and the value of VIF was within the acceptable range which showed no collinearity. Therefore, there was no correlation between GA and other risk factors of anemia in our study which means the risk factors of anemia were not changeable based on differences in gestational age.

---

## [Editor Report · Decision Letter 2]

16 Oct 2019

Determinants of Anemia among Pregnant Women Attending Antenatal Care in Horo Guduru Wollega Zone, West Ethiopia: Unmatched case-control study

PONE-D-19-15457R2

Dear Dr. Mengist,

We are pleased to inform you that your manuscript has been judged scientifically suitable for publication and will be formally accepted for publication once it complies with all outstanding technical requirements.

With kind regards,

Mary Glover-Amengor

Academic Editor

PLOS ONE
---

## [Editor Report · Acceptance letter]

21 Oct 2019

PONE-D-19-15457R2 

Determinants of Anemia among Pregnant Women Attending Antenatal Care in Horo Guduru Wollega Zone, West Ethiopia: Unmatched case-control study 

Dear Dr. Mengist:

I am pleased to inform you that your manuscript has been deemed suitable for publication in PLOS ONE. Congratulations! Your manuscript is now with our production department. 

With kind regards,

on behalf of

Dr. Mary Glover-Amengor 

Academic Editor

PLOS ONE